# Enhanced magnetic spin–spin interactions observed between porphyrazine derivatives on Au(111)

Jie Hou[1,2], Yu Wang[1,2], Keitaro Eguchi[3], Chihiro Nanjo[3], Tsuyoshi Takaoka[1,2], Yasuyuki Sainoo[1,2], Ryuichi Arafune[4], Kunio Awaga[3] & Tadahiro Komeda [2✉]

Magnetic molecules are of interest for application in spintronic and quantum-information processing devices. Therein, control of the interaction between the spins of neighboring molecules is the critical issue. Substitution of outer moieties of the molecule can tune the molecule–molecule interaction. Here we show a novel spin behavior for a magnetic molecule of vanadyl tetrakis (thiadiazole) porphyrazine (abbreviated as VOTTDPz) adsorbed on Au(111), which is modified from vanadyl phthalocyanine (VOPc) by replacing the inert phthalocyanine ligand with a reactive thiadiazole moiety. The magnetic properties of the molecules are examined by observing the Kondo resonance caused by the screening of an isolated spin by conduction electrons using scanning tunneling spectroscopy. The Kondo features are detected at the molecule whose shape and intensity show site-dependent variation, revealing complex spin–spin interactions due to the enhanced interaction between molecules, originating from the functionalization of the ligand with a more reactive moiety.

---

[1] Department of Chemistry, Graduate School of Science, Tohoku University, Aramaki-Aza-AobaAoba-Ku, Sendai 980-8578, Japan. [2] Institute of Multidisciplinary Research for Advanced Materials (IMRAM, Tagen), Tohoku University, 2-1-1 KatahiraAoba-Ku, Sendai 980-0877, Japan. [3] Department of Chemistry and Research Center for Materials Science, Nagoya University, Chikusa-ku, Nagoya 464-8602, Japan. [4] Research Institute of Electrical Communication, Tohoku University, 2-1-1 KatahiraAoba-ku, Sendai 980-8577, Japan. ✉email: tadahiro.komeda.a1@tohoku.ac.jp

Magnetic properties of adsorbed molecules on surfaces have attracted attentions recently with a rising interest for the spintronic and quantum-information processing devices, where a realization of the spin entanglement is one of key issues[1]. As a standard molecule, many studies have been published for the spin properties of the metal phthalocyanine (Pc) molecules[2]. However, the number of reports for the phenomenon caused by their spin–spin interaction is limited. This is partially because a strong molecule–molecule and molecule–substrate interaction cannot be expected for the Pc molecules, whose perimeter is terminated with inert C-H species[3,4]. There have been continuous efforts towards the formation of a stronger molecule–molecule interaction for Pc family molecules, which include a substitution of the benzene rings with other moieties[5]. Stuzhin and colleagues[6,7] first synthesized tetrakis(1,2,5-thiadiazole) porphyrazines and corresponding MTTDPz derivatives (M = Mn(II), Fe(II), Co(II), Ni(II), Cu(II) and Zn(II)). A thiadiazole is a five-membered compound containing nitrogen and sulfur atoms, which is more reactive compared with the benzene ring of the Pc group. They pointed out extensive $\pi$ electrons delocalize over the skeleton with considerable overlap between adjacent 1,2,5-thiadiazole rings. Afterwards, Suzuki and colleagues[4,8,9] showed that a significant intermolecular interaction is formed between these molecules, which is due to a strong side-by-side electrostatic attractive force of S···N between the adjacent MTTDPz molecules. They demonstrated that the existence of peripheral heterocyclic rings with $\pi$ electron deficient results in the delocalization of the electronic charge into the whole skeleton of tetrakis (thiadiazole) porphyrazine (TTDPz)[6,7]. We previously reported the synthesis and structural analysis of

the vanadyl TTDPz (VOTTDPz) molecule[10]. It was confirmed that the bulk crystal shows two polymorphs, α- and β-forms; the former is a one-dimensional (1D) and the latter is a two-dimensional (2D) crystal. For magnetic properties, the molecule has S = 1/2 spin and we found a ferromagnetic (FM) ordering between molecules in the α-form.

In this study, we investigate VOTTDPz molecule film grown on Au(111) surface using scanning tunneling microscopy and spectroscopy (STM/STS) focusing on its magnetic properties. Compared with the cases for the metal Pc molecules, compact lattices in the film are formed through a strong interaction between the thiadiazole groups. We analyze the magnetic properties by examining the Kondo resonance, which is caused by a screening of an isolated spin by conduction electrons[11] and appears in STS near the Fermi level. Two types of orderings of the molecules in the films are observed. First one, phase I, is composed of flat-lying molecules, where the VO group is directing towards the vacuum side. The Kondo peak was observed at both the center and the ligand position in the STS spectra. The spectrum obtained at the latter position shows an unusual increase in intensity when outer magnetic field is applied. An intriguing spatial variation of the Kondo peak, which is the crossover from the Kondo peak to Fano dip, is observed within a single molecule, the latter of which is caused by an interference between the tunneling current with the paths of tip–substrate and tip–molecule–substrate. A strong coupling between the thiadiazole group and the substrate forms the Fano dip in the former case, whereas a weak coupling between them provides a peak in the latter case. It demonstrates that the Fano shape analysis can be a tool to examine the local chemical environment. In the second type of the film (phase II), a VO-up molecule showing a flat-lying bonding and a VO-down molecule with a tilted configuration appear in an alternative manner. The spin-polarized density functional theory (DFT) calculation shows that, even though the spin remains on all molecules, there are two types among the VO-down molecule in terms of the spin coupling with the neighboring molecules. First has a FM coupling with the flanking two flat-lying molecules and the other has an antiferromagnetic (AFM) coupling with them. A clear Kondo peak appears at the former molecule but it disappears when measured at the latter position. This phenomenon is explained by the competition between the Kondo resonance and Ruderman–Kittel–Kasuya–Yosida (RKKY) interaction between the neighboring spins, which is enhanced with the presence of the reactive thiadiazole group.

## Results and discussion

**VOTTDPz molecule and its film on Au(111).** Figure 1a shows a model of a VOTTDPz molecule used in this report. The shape is similar to a vanadyl Pc molecule, but a VOTTDPz molecule contains four thiadiazole groups (specified by a red circle in the figure) whose perimeters are terminated by S (yellow) and N (blue) atoms. There are two other types of N atoms: one is at the apex of the pyrrole group (marked by a brown circle) and the other is bridging pyrroles (green circle). The VO group is directing out of the molecular plane, which provides an additional freedom for the adsorption configurations: VO-up and VO-down cases. As discussed previously[6,7], stronger interactions of molecule–molecule and molecule–substrate are expected through thiadiazole groups. When the molecules are deposited onto a clean Au(111) surface whose temperature is kept at room temperature, we see the formation of a film that is composed of islands with well-ordered lattices of molecules (see Fig. 1b). We notice that the islands that are marked by I and II have different orderings of the molecules even though both of the islands have a monolayer height.

**a**

**b**

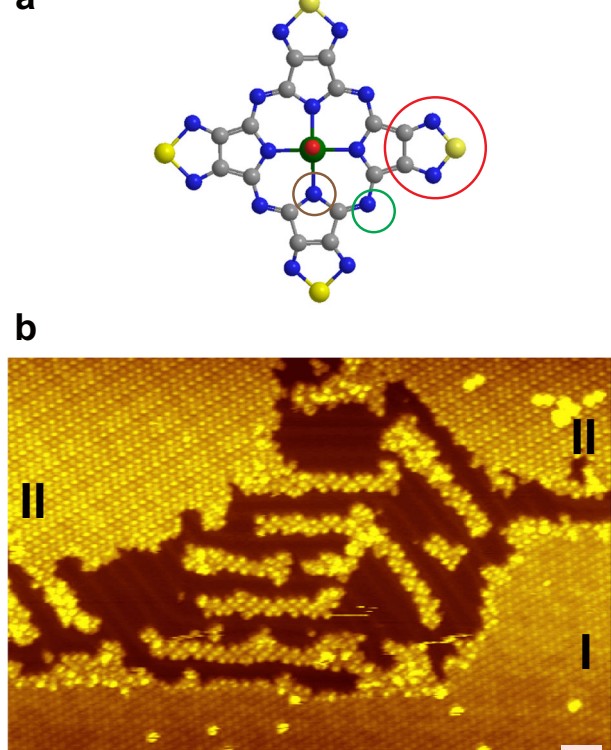

**Fig. 1 Structural model and STM image of the VOTTDPz film. a** Optimized structure of VOTTDPz. Color code: yellow, S; bright blue, N; cyan, C; green, V; red, O. **b** STM topographic image of the VOTTDPz monolayer film (size 80 × 54 nm², white scale bar 5 nm, $V_s = -0.8$ V, $I_t = 160$ pA). Two types of molecular arrangement of films are marked by **I** and **II**.

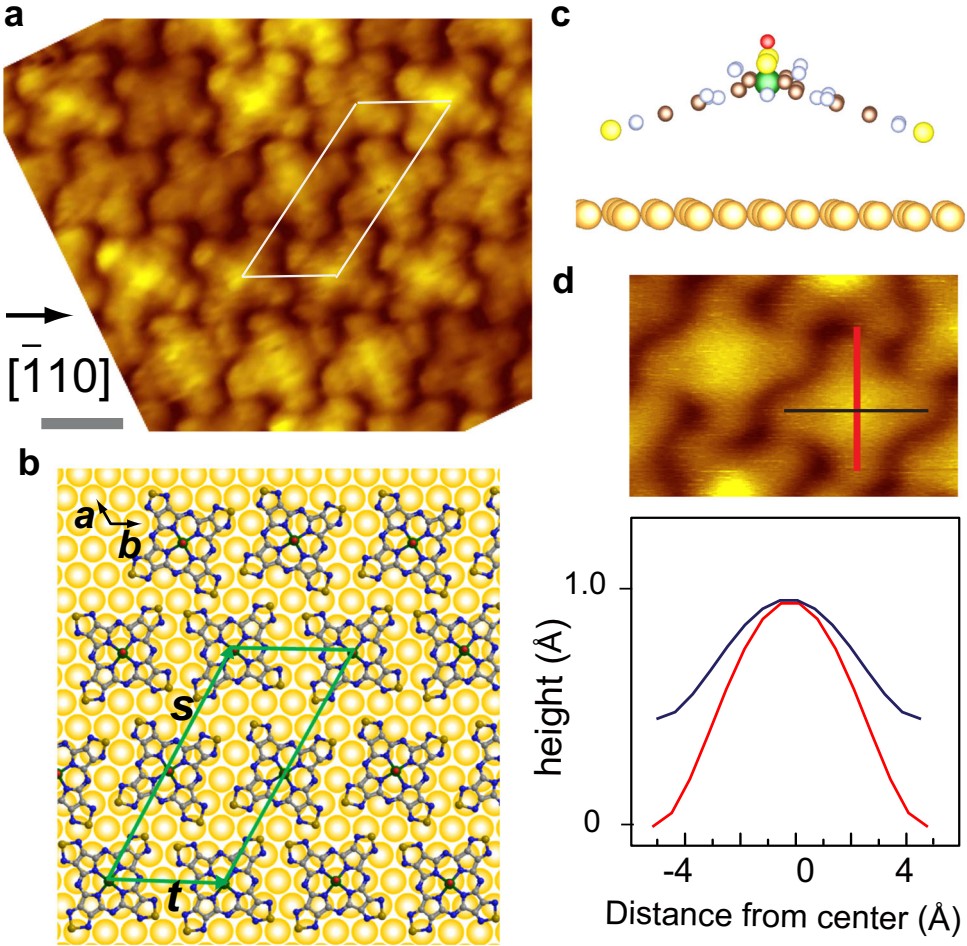

**Fig. 2 STM image and model structure of phase I film. a** Magnified STM image of phase I film of VOTTDPz monolayer film ($V_s = -0.8$ V, $I_t = 160$ pA). Crystallographic direction of Au substrate is indicated by an arrow. Length bar (grey line) is 1 nm. **b** Top view of the adsorption model. Green parallelogram (unit vectors **s** and **t**) corresponds to the unit cell containing two molecules, corresponding to the one in **a** with white lines. **c** Side view of the same model of **b**. **d** Height profile vs. horizontal distance along two symmetry lines in the lower panel, whose lines are superimposed in the upper topo image with the same color.

**Lattice structure of VOTTDPz film: phase I**. We first analyze the lattices of the films of the phase I. In Fig. 2a, we show a magnified image of the phase I island where the [−110] direction of Au(111) substrate is aligned to the horizontal direction (preliminary analysis was published[12]).

The unit cell can be expressed similar to the parallelogram shown in Fig. 2a, which contains two molecules rotated with each other. After placing the molecules in the corresponding positions, we executed the DFT calculation using the Vienna Ab initio Simulation Package (VASP) code for the structural optimization. We show the optimized structure in Fig. 2b with the unit vectors **s** and **t** indicated by green lines (the coordination is shown in Supplementary Note 1), which can be expressed using the ones of the substrate, **a** and **b**, similar to the matrix formula (1).

$$\begin{pmatrix} s \\ t \end{pmatrix} = \begin{pmatrix} 9 & 9 \\ 0 & 4 \end{pmatrix} \begin{pmatrix} a \\ b \end{pmatrix} \qquad (1)$$

As illustrated in Fig. 2b, the directions of the symmetry line of the molecules are aligned to the close-packed directions of Au (111).

In addition, we can see an intriguing molecule deformation in the side view shown in Fig. 2c. Two S atoms among four, which exist at the end of the thiadiazole group, bent towards the substrate, whereas the other two are bent towards the vacuum.

We can detect this bent configuration of the molecule in the topo image by measuring the cross-sectional height variation in the lateral direction. In Fig. 2d, we show the cross-sectional height information of the molecule in the phase I film along the two symmetry lines, which are indicated by the colors of red and black superimposed on the topographic image. The height profile illustrated in Fig. 2d suggests that the heights of the lobes along these two lines are not identical. Rather, the one along the red line is bent towards the vacuum side compared with that along the black. Thus, it is in the twofold symmetry condition, even the difference is rather small. This supports the optimized structure shown in Fig. 2c in which the bending angles of the lobes along the two symmetry lines are not identical.

**Lattice structure of VOTTDPz film: phase II**. The ordering of the molecules in the phase II film is more complicated if compared with that of the phase I, but it also shows a commensurate structure. The magnified image and the temporal models are shown in Fig. 3a, b, respectively. These two are aligned with the common crystallographic directions of the substrate, which are depicted at the lower-right corner of Fig. 3a with larger circles together with the two unit vectors of the substrate marked by **a** and **b**. The unit vectors of the superstructure of the molecule film

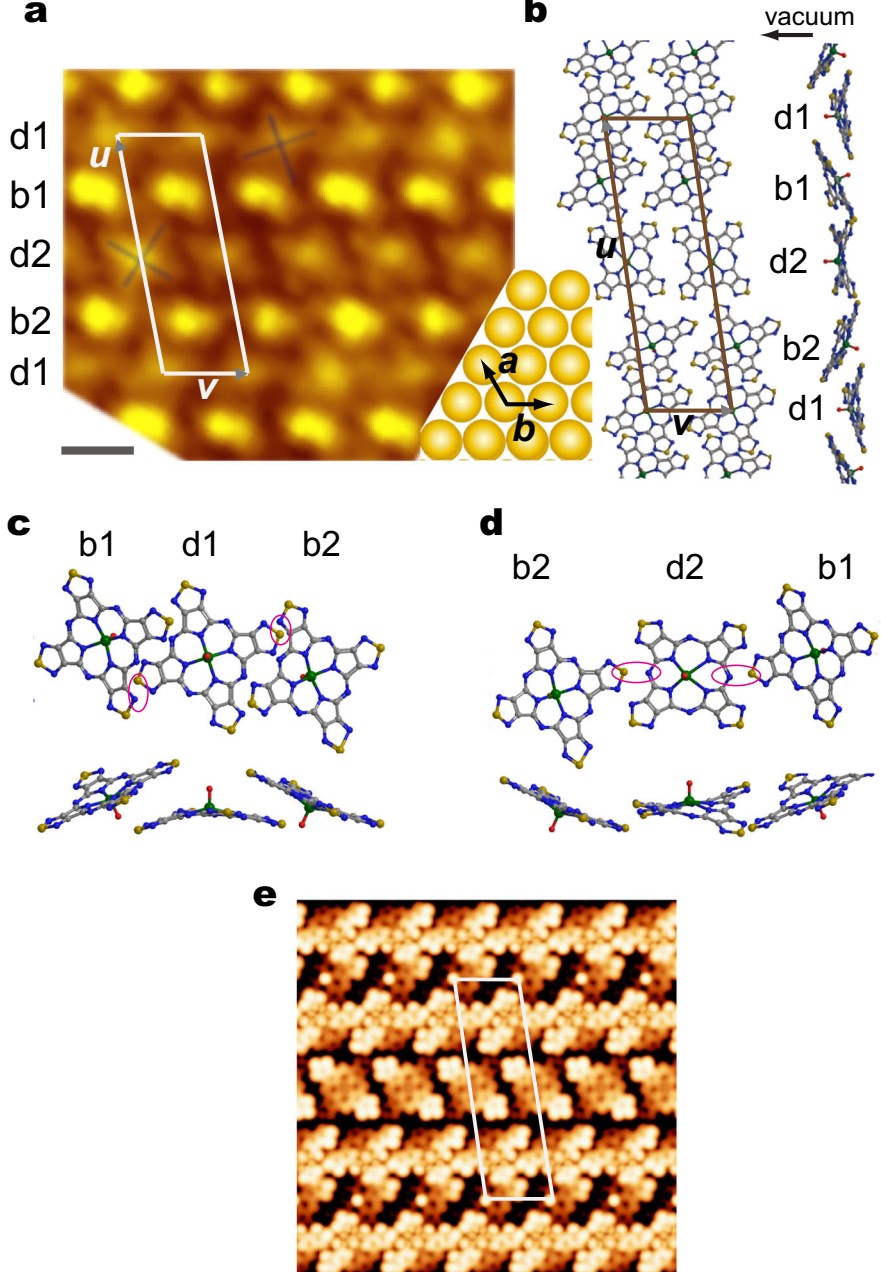

**Fig. 3 STM image and model structure of phase II film. a** Magnified STM image of phase II film of VOTTDPz (gray scale bar = 1 nm, $V_s = -0.8$ V, $I_t =$ 160 pA). Crystallographic directions of Au(111) are shown in the lower-right corner. White parallelogram correspond to the unit cell containing four molecules, b1 and b2 (d1 and d2) are bright (dark) molecules. **b** Optimized structure of the phase II model. Top view and side view are provided, together with unit cell and unit vectors corresponding to the one of **a**. **c**, **d** Magnified models of **b** for the b1–d1–b2 part (**c**) and for the b2–d2–b1 part (**d**). Both top view and side views are provided. **e** Simulated STM image for the image of **a**, in which the unit cell is indicated by white lines.

can be described using the following matrix,

$$\begin{pmatrix} u \\ v \end{pmatrix} = \begin{pmatrix} 16 & 6 \\ 0 & 4 \end{pmatrix} \begin{pmatrix} a \\ b \end{pmatrix} \quad (2)$$

where **u** and **v** are the unit vectors of the molecular film, illustrated in Fig. 3a, b. As can be seen in Fig. 3a, the unit cell contains four molecules in which b1 and b2 are bright molecules, and d1 and d2 are dark ones.

In d1 and d2, we see a cross-like feature as the inner structure, which is marked by dark crosses in the figure. These features are those assigned to the flat-lying molecule with the VO-up configuration in the previous section. Thus, we start with placing

the flat-lying molecules with the azimuthal rotation angles deduced from the STM images. With the manner of the presentation of Fig. 3, all molecules in a row have an identical bonding configuration.

On the other hand, the bright molecules of b1 and b2 show no cross-like features. For the reason of the absence of the cross shape, we consider a tilted configuration of the molecule with following three reasons.

First, if we consider a model of the molecule lattice and place the dark molecules of d1 and d2 according to the STM image, it is impossible to place the bright molecules of b1 and b2 as flat-lying ones without a steric repulsion problem, because the lattice size is smaller than that for the phase I.

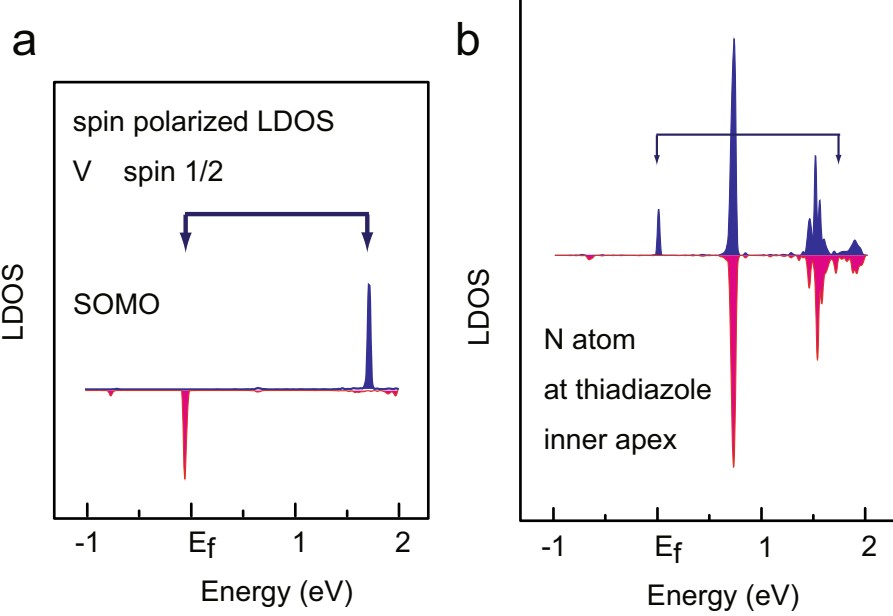

**Fig. 4 Spin-resolved LDOS vs. energy plots.** LDOS calculated for V atom (**a**) and for N atom at the pyrrole group (**b**).

Second is the disappearance of the cross-like feature in the STM image. Without an additional atom deposition or the presence of contaminations, the flat-lying VOTTDPz should show a symmetric shape in the STM image. It can be speculated that the tilted configuration diminishes the cross-like feature due to an asymmetric bonding configuration.

Third is a comparison with the bulk crystal structure of this molecule examined in our previous reports[10]. As previously stated, we reported the bulk crystal structures for the α- and β-phases that correspond to the 1D π-stacking and the 2D π-assembly, respectively, based on our single crystal X-ray analysis. The former appears in a needle-like crystal, whereas the latter is observed in a layer-like one. Here we compare the phase II structure with the β-phase due to its 2D-like nature. As expressed in the Supplementary Note 3 and Supplementary Fig. 1, the 2D π-network in the β-phase forms an assembly in which VO-up and VO-down molecules appear in an alternative manner. Even though the centers of the molecules are placed in a 2D plane, the neighboring VO-up and VO-down molecules have a buckled configuration (for side view image, see Supplementary Fig. 1). Thus, it can be speculated that a buckled bonding between two neighboring molecules with reversed VO directions corresponds to the one observed in the bulk β-phase. As both of the flat-lying molecules of d1 and d2 have VO-up configuration, the ones in the tilted positions should have VO-down configuration if it follows the bulk ordering. With these three reasons, we believe that the bright molecules have a bent configuration. The tilted configuration of VO-down molecule can also be deduced from a physical consideration, because the system can gain energy by tilting the molecule to make a bonding between the S atom and the substrate.

After placing b1 and b2 at the positions observed in the topo image with the tilted configuration, we make a temporal model that is optimized using VASP calculation. The optimized model is shown in Fig. 3b, where both of the top and side views are depicted. As obvious in the side view, the flat-lying and tilted molecules appear in an alternative way. For the flat-lying molecules, two of the four thiadiazole groups seem to have a strong bonding with Au substrate and the others are bent towards the vacuum (the coordination is shown in Supplementary Note 2).

Magnified models around the flat-lying molecules, d1 and d2, are shown in Fig. 3c, d, respectively. The molecule d1 is flanked by b1 in the left- and by b2 in the right-hand side, both of which are tilted towards the vacuum away from d1. On the other hand, b1 and b2 molecules bent downwards from d2. Although both d1 and d2 have a flat-lying configuration, their interactions with neighboring molecules are different. For d1 molecule, N atom bridging pyrroles (green circled in Fig. 1a) has the nearest-neighbor distance to the b1 and b2 molecules, which are marked by red circles. The molecule d2 is bonded with neighbors with the thiadiazole marked in Fig. 3d.

The evidence of the canted configuration of the molecule in the second phase can be obtained by comparing the topo image with a simulated STM image based on the optimized structural model of this molecule. In Fig. 3e, we illustrate the simulated STM image, which is calculated using the Terssof–Herman method based on the electronic configuration obtained by the DFT calculation with the VASP code. In the image of Fig. 3e, we superimposed the unit cell same as that of Fig. 3a. The simulated images of the flat molecules of d1 and d2 show four stems, although two are highlighted reflecting the bent configuration causing the twofold symmetry. However, the molecules of b1 and b2 have three lobes highlighted, which appear as a ball-like shape in the topo image. The similar combination of the two types of the molecule appearance in the STM image and the simulated image supports the structural model.

**Spin property of VOTTDPz film**. Next, we switch to a discussion of magnetic properties of the adsorbed VOTTDPz molecules. First, we calculate the spin-polarized local density of states (LDOS) for a molecule in the phase I film with using VASP code. The results are shown in Fig. 4. The LDOS calculated for the V atom shows a spin-polarized 3$d$ state (mainly contributed from $d_{x^2-y^2}$) showing a clear singly occupied molecular orbital (SOMO) state. At the same time, the spin is delocalized in the porphyrazine of the ligand. The LDOS for the N atom at a pyrrole is shown in Fig. 4b, in which we see a SOMO level just below the Fermi level and the corresponding state with the reverse spin at the position of the arrow. Thus, the spin is delocalized over the

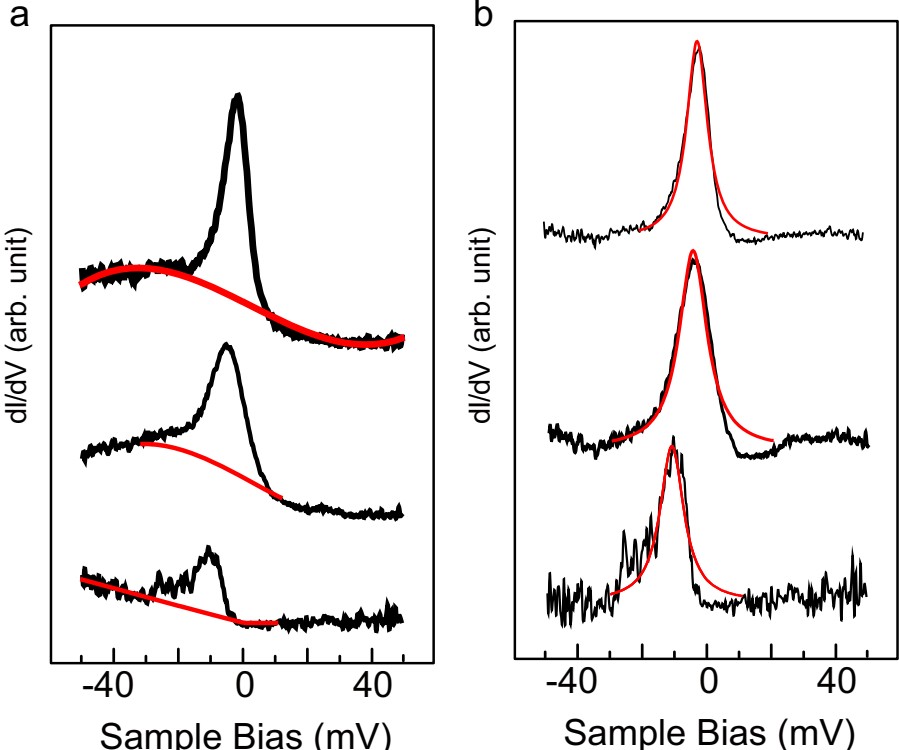

**Fig. 5 Magnetic-field-dependent dI/dV curves near zero bias. a** dI/dV obtained at the VOTTDPz molecule of phase I film, where the tip was place at the ligand position. The estimated backgrounds are shown by red lines. Magnetic fields along the out-of-plane direction 0, 3, and 5 T are applied from bottom to top. **b** Same as **a**, except that the background is subtracted. The fitting results are shown by red lines (see text).

ligand and not only localized at the VO position. A similar result was reported by our group from the calculation using Gaussian code[10].

**Kondo resonance of VOTTDPz molecule in phase I film**. Here we examine the magnetic properties of the molecules by detecting the Kondo resonance with measuring STS near the Fermi level. In Fig. 5, we show the variation of the STS spectra with the application of different magnetic fields.

We first show the STS curves and the fitted background curves in Fig. 5a, the latter of which are indicated by the red lines. For the plots in Fig. 5a, the heights of the spectra are normalized so that the intensities of the background curves at the Fermi level ($G_0$) are equivalent. We define the height of the zero bias peak (ZBP) to be that after the subtraction of the background and define as $\Delta G$. The height after the normalization by the $G_0$ ($\Delta G/G_0$) can be read as 0.38, 0.86, and 1.40 for $B = 0$, 3, and 5 T, respectively.

Apparently, the peaks become stronger and more symmetric, and the peak positions shift towards the Fermi level as the applied magnetic field of $B$ increased from 0 to 5 T.

The sharp feature near the Fermi level originated from the magnetic adsorbate is often related to the Kondo resonance. Thus, as has been demonstrated in previous reports, the spectra after the background subtraction were analyzed with a model of the Fano resonance, which is expressed using the following equation[13,14]:

$$dI(V)/dV \propto \frac{\rho_0(\varepsilon + q)^2}{\varepsilon^2 + 1}, \varepsilon = \frac{eV - \varepsilon_0}{\Gamma} \tag{3}$$

where $q$ is the Fano parameter, $\varepsilon_0$ is the peak position, and $\Gamma$ is the resonance width. The shape of the spectra changes according to the parameter $q$, where large and small $q$ correspond to the

cases in which the majority of the tunneling path is through the molecule and directly to the metal, respectively. When $q \gg 1$ ($q \sim 1$), the spectrum becomes a simple peak (dip) whose full width at half maximum is $2\Gamma$.

The results of the fitting using Eq. (3) are shown by the red lines in Fig. 5b, in which the $\varepsilon_0$ changes from $-12.0$ mV for $B = 0$ T towards $-5.5$ mV for $B = 3$ T and $-2.5$ mV for $B = 5$ T. The peak widths of the one for $B = 0$ T and for $B = 5$ T are 9.0 mV and 7.8 mV, respectively.

We check the previous reports of Kondo features studied for metal Pc molecules. Those can be divided into two groups as follows: (1) Kondo features originated from the spin of a metal $d$ electron and (2) those induced from the spin of ligands. For the case of (1), the Kondo features tend to appear as a complex shape instead of a symmetric peak. These include FePc[15,16], MnPc[17–20], and hexadeca-fluorinated iron Pc[20], where a dip or staircase-type Kondo resonance is reported. Examples of the (2) can be seen in the report for the molecules of NiPc and CuPc[21]. In there, sharp symmetric peaks are observed as the ZBP, which are assigned to the features originated from the $\pi$-radical created by a charge transfer from the substrate to the molecule. A similar symmetric sharp Kondo peak is also observed for TbPc$_2$ double-decker molecule adsorbed on Au(111). The TbPc$_2$ molecule has a $\pi$-radical due to the trivalent nature of the Tb atom in the neutral state, which produces a symmetric Kondo peak[22–25].

Liu et al.[19] observed changes of the Kondo feature by sequentially removing hydrogen atoms from a MnPc molecule (dehydrogenated states are characterized by the missing number of H atoms, which is controlled from one to eight). By this manipulation, the step-like Kondo feature of MnPc smoothly changes to Kondo peak. The observed phenomenon is attributed to the shift of the spin polarization towards the lobe position, which may arise from a charge transfer within the molecule after

the dehydrogenation. A similar spin-state variation with chemical change was observed by Perera et al.[26]. In there, the DFT calculation executed for the dehydrogenated MnPc molecule shows that the spin polarization shifts from the $Mn^{2+}$ ion to the benzene ring after dehydrogenation.

For the system of this report, the spin polarization is distributed both at the V atom and at the ligand. In addition, the symmetric Kondo peak is a hallmark of the Kondo resonance originated from the ligand. We consider a model that the spin distribution weight is shifted from the V atom to the ligand with the B application due to the shift of the energy positions with magnetic field. However, Liu et al.[19] showed that the energy position of the staircase feature, which was confirmed as the Kondo resonance by examining the splitting in the magnetic field, is aligned with the Fermi level. This shows a discrepancy from our observation.

Instead, the change of the ZBP can be accounted for with a model of the crossover from the Kondo regime to the mixed valence (MV) state. With the Anderson model, the Kondo resonance is modeled with a singly occupied electronic state. Several regimes are considered depending on the energy position $E_0$ and the peak width $\Delta$ of the singly occupied state; when $E_0/\Delta \gg 1$ it is grouped as the Kondo state, whereas when the broadened state is overlapped with the Fermi level ($E_0/\Delta < 1$) it is grouped as the MV state.

The spectral functions, which can simulate the tunneling spectra, are calculated theoretically in several reports. Li et al.[27] calculated a case where the magnetic atom is placed on the graphene substrate. Also, we can see similar crossover simulation for the rare-earth inter-metallics[28], together with an experimental report[29].

In the Kondo regime the spectral function shows a peak located at the Fermi level with a sharp peak width, whereas in the MV regime the peak position is off the Fermi level and the peak width is wide. The former and latter features are what were observed for $B = 5$ T and $B = 0$ T in this experiment, respectively. Thus, we consider that a crossover from the MV regime to the Kondo resonance regime is induced by the magnetic field. An example of the switching from the Kondo region to the MV regime can be seen in the carbon nanotube field-effect transistor device. By tuning the gate voltage, the relative position of the spin center in reference to the Fermi level can be tuned. The results showed characteristic features of the conductance in both regimes[30].

For the mechanism of the crossover from the MV regimes to the Kondo regime with B application, we should consider mechanical distortion of the molecule with the presence of the magnetic field, which can induce the reduction of the coupling between the molecule and the substrate. However, much more detail should be examined in the future.

One might wonder why the Zeeman splitting is not obvious for the spectra with the application of $B = 5$ T. Park et al.[31] demonstrated that when the outer magnetic field is applied, the Kondo peak is split into two peaks with the energy separation of twice the Zeeman energy. However, compared with this case, the original broadening of the Kondo feature is larger in the experiment shown in this report, which makes it difficult to see the splitting. In our previous report, we simulated the changes in the peak by convoluting two Lorentzian peaks with the width of 6.5 meV and separated by 1.8 meV (two times the Zeeman splitting energy at 8 T) around the Fermi level into a single peak. Due to the original peak width of ~6.5 mV, the splitting was not obvious, except for a slight broadening of ~1 meV. As a similar broadening in the original peak, the Zeeman splitting is not obvious even with the application of the outer magnetic field.

Next, we examine a spatial variation of the Kondo feature. As a sharp and strong Kondo peak can be observed with an application of external magnetic field, STS spectra shown here are obtained with $B = 3$ T condition. The results are summarized in Fig. 6 and the tip positions for the STS measurements are marked on the STM image of Fig. 6a, which are on the four thiadiazole groups of a single molecule.

As already discussed, our DFT calculation predicts a bent configuration for the thiadiazole groups, where two are bent towards the surface and the other two towards the vacuum. This is illustrated in the optimized model of Fig. 6b, where the thiadiazole groups on the line connecting I and V are bent towards the vacuum and those on the A–D line are bent towards the substrate. Experimentally, STS spectra near the Fermi level show peaks on the line of I–V and dips on the line of A–D.

Such a dip feature observed for II and IV has been discussed previously and understood with the interference between the tunneling electrons with the paths of (1) direct tunneling from the tip to the substrate and (2) tunneling through a molecule[32]. The interference of the two paths causes a Fano dip[13]. The Fano shape is scaled with the interaction between the spin center and the substrate, stronger of which causes the Fano dip and weaker one corresponds to a peak.

Using Eq. (1), we executed the fitting whose results for the plots of IV and B are indicated by red lines superimposed on the original plot of Fig. 6c. The fit parameters for B are $q = -0.08 \pm 0.01$ and $\Gamma = 7.7 \pm 1.5$ mV, and those for IV are $q = 467 \pm 1$ and $i = 8.6 \pm 1.7$ mV.

The crossover of the Kondo peak and Fano dip observed in the experiment can be explained by the change of the coupling between the molecule and the substrate. The interaction between the two is determined by the local coupling between the functional group of the thiadiazole and the Au substrate. Thus, $q \sim 1$ is expected for the lobes on the line of I–IV. However, this assumption stands only if the variation of local chemical environment manifests itself in the Fano shape of the Kondo resonance with a sub-molecule resolution in a single molecule.

We previously reported such a change of the Kondo peak for $TbPc_2$ molecule[24]. The Kondo temperatures measured at the four pyrrole groups of the top Pc ligand show different values within a single molecule, which is explained by the tilt of the top Pc that changes the distance of pyrrole to the substrate[24]. The results shown in this experiment suggest that the Fano resonance appearing in Kondo peak is sensitive to the local chemical environment with a submolecular spatial resolution, which might be used as a chemical analysis technique.

**Spin interaction in phase II film**. Here we examine the Kondo features observed for the phase II film. We show STS spectra in the energy region near the Fermi level in Fig. 7, which are obtained at the ligands of the four molecules in the unit cell specified in Fig. 3. Again b1 and b2 are the molecules that appear as bright molecules, whereas d1 and d2 are dark ones. Magnetic field of 3 T is applied to obtain the stronger Kondo peak. Interestingly, in the spectra obtained at b2, the Kondo peak disappears and only a broad shallow dip can be seen, whereas the other three show a clear Kondo peak.

The appearance and disappearance of Kondo peak depending on the measurement site for the phase II film should be examined through a comparison with a theoretical simulation of the spin distribution. In Fig. 8, we show spin polarization distribution calculated with spin-resolved DFT calculation using VASP code for the model of Fig. 3b, in which the majority and minority spin components are colored with green and red, respectively. We see large spin polarization at the V and O atoms, and less intensive polarization at the thiadiazole ligand part.

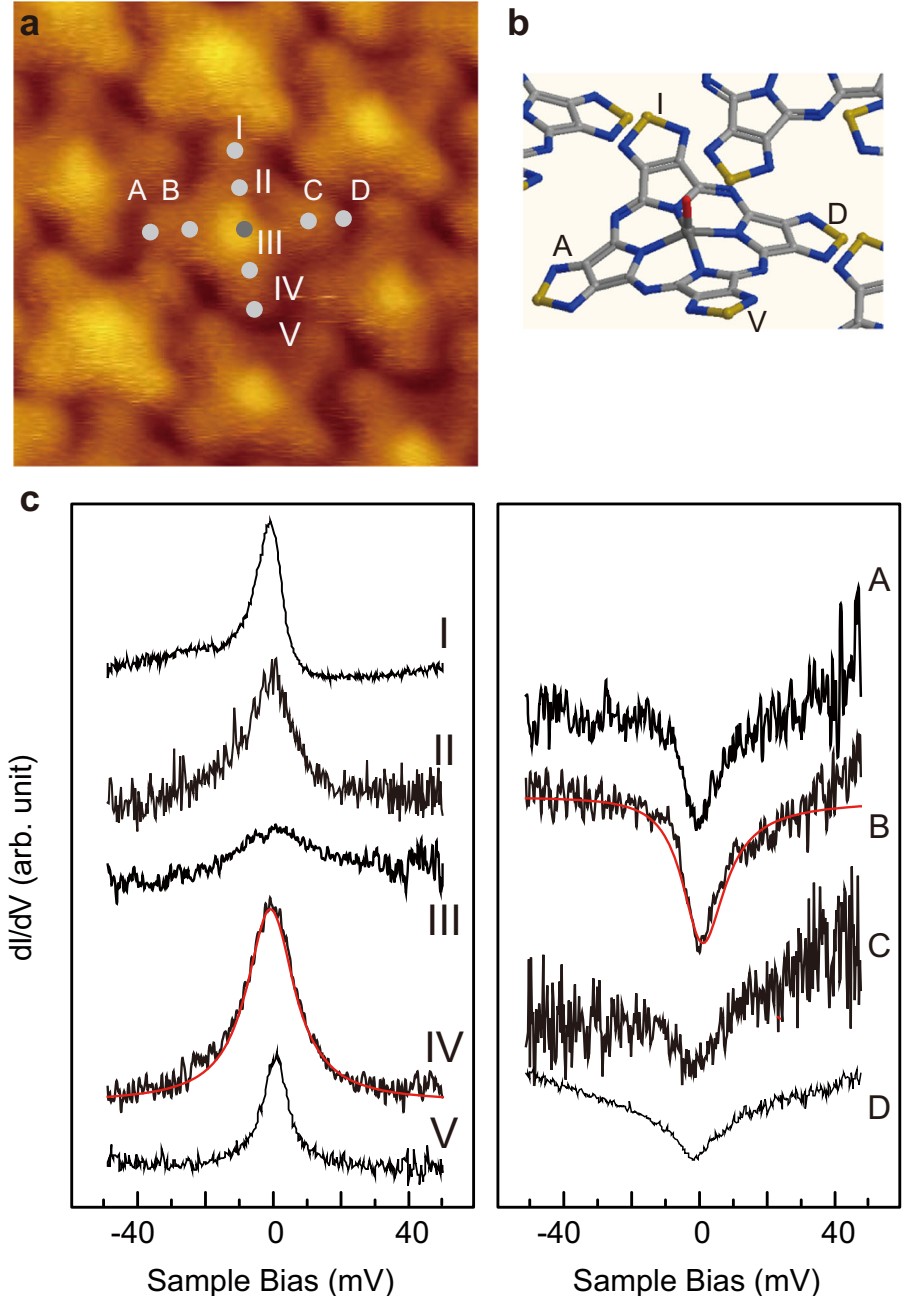

**Fig. 6 Kondo peak variation with inner-molecule positions for phase I film. a** Magnified STM image of phase I film, indicating the positions for STS measurements. **b** Model of the target molecule illustrated in a three-dimensional manner. Note pyrroles of I and III (II and IV) are bent away from (towards) the substrate. **c** Kondo resonance features of VOTTDPz molecule at the positions specified in **a**. $B = 3$ T were applied to clarify the spectra.

We should note that the spin polarization remains in all molecules, which seems inconsistent with the absence of the Kondo feature at b2 molecule. However, it should be noted in Fig. 8 that b2 molecule has a reverse spin direction with respect to the other three (b1, d1, and d2). Consequently, if we examine bright molecules, b1(b2) has the FM (AFM) coupling with neighboring molecules of d1 and d2. The difference of b1 and b2 molecules in terms of the spin coupling between the neighboring molecules could explain the difference of the Kondo resonance observed at b1 and b2.

For the coupling mechanism between the spins of neighboring molecules, we should exclude the magnetic dipole coupling or direct magnetic coupling due to the large distance between the V atoms of neighboring molecules. In addition, a direct interaction

of the electronic states between the neighboring molecules can cause such an effect. However, it should be noted that the two molecules of b1 and b2 are equivalent in terms of the local structure including the neighboring molecules of d1 and d2, which should be followed by the identical molecule–molecule interaction Thus, the presence and the absence of the Kondo state at b1 and b2 cannot be explained solely by this effect.

Instead, we shall consider an interaction between spins through conduction electrons; i.e., RKKY) interaction. Although the Kondo effect screens the impurity spin and makes the system nonmagnetic, RKKY makes the magnetic moment stabilized. The studies of the two impurity spin system showed that modifications of the spectrum shape of the Kondo resonance by the presence of RKKY interaction. It has been reported that an RKKY

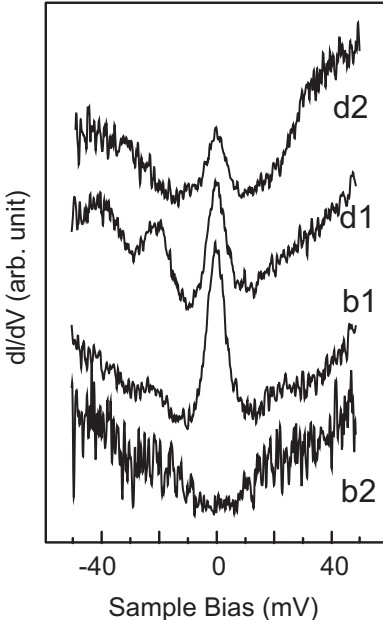

**Fig. 7 Variation of Kondo peaks for the phase II film.** The d*I*/d*V* spectra near the Fermi level obtained for the four types of the molecules in the phase II film. The marks of b1, b2, d1, and d2 are corresponding to molecules specified in Fig. 3.

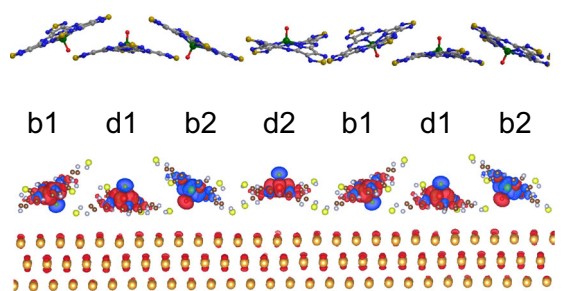

**Fig. 8 Spin polarization distribution of phase II film.** The distribution of the spin polarization for phase II film is mapped with color contrast. Red and blue colors represent the majority and minority spin density, respectively. Refer to Fig. 3 for molecule models.

coupling broadens the width and reduces the intensity of the Kondo peak if the coupling between the spins is AFM, whereas it reduces the width and enlarges the peak intensity for the FM-coupling case[33–36]. Tsukahara et al.[16] demonstrated that the RKKY interaction effects can be observed for the FePc islands. They observed that the Fano-shape Kondo resonance originated from Fe 3*d* is broadened and split into multiple features, which is attributed to the AFM coupling between FePc molecules in the film. We believe this effect is more enhanced for the system shown in this study, as the molecule–molecule spacing is smaller for the current system due to the compact thiadiazole ligand. The difference of the shape of the Kondo resonance observed for the b1 and b2 molecules should be caused by the difference of the interaction with neighboring molecules; b1 has a FM coupling with neighbors, whereas b2 has a AFM coupling. According to the previous studies for the Kondo resonance and RKKY coupling, the former should have a sharp peak width and large peak intensity, whereas the latter should have the broad and weak peak intensity. These expected behaviors go along with the observed results. Similar spin–spin interaction effects on the Kondo resonance have been reported for several systems[20,37]. We want

to stress that the coupling between the Kondo resonance and RKKY is emphasized in the current system due to the smaller molecule–molecule distance.

We discuss an additional technique to verify such an FM/AFM magnetic ordering. It has been demonstrated that the X-ray magnetic circular dichroism (XMCD) is a powerful technique to analyze the magnetic behavior of the molecule film. We show preliminary result of the XMCD comparing the films of the VOPc and VOTTDPz molecules in Supplementary Note 4 and Supplementary Fig. 2. The XMCD results indicate that the magnetization appeared on the VOTTDPz film is much weaker that that obtained from the VOPc film. The result can be interpreted by a model that, compared with the paramagnet nature of VOPc, the appearance of the AFM magnetic ordering between molecules in a part of the film (corresponding to the phase II part) should correspond to the decrease of the magnetization.

In summary, we investigated VOTTDPz molecular films grown on a Au(111) surface using STM and STS, focusing on its intriguing magnetic properties. We examine the magnetic properties by examining the Kondo resonance in STS near the Fermi level. Two types of ordering of the films are observed. The first film, phase I, is composed of flat-lying molecules, where VO group is directing to the vacuum side. A spin-resolved DFT calculation reveals that a VOTTDPz molecule has a spin both at the V atom and delocalized at ligand positions. Experimentally, the Kondo peak is observed both at the V position and the ligand position in STS spectra, the latter of which increases in intensity when outer magnetic field is applied. The spatial variation of the Kondo peak is observed within a single molecule, which corresponds to the crossover from Kondo peak to Fano dip. The DFT calculation shows that, among four thiadiazole groups of a single molecule, two of them have a strong bonding with Au substrate at S atom and the others are bent towards the vacuum side. The strong coupling between the former thiadiazole group and the substrate forms the Fano dip and the weak coupling of the latter provides a peak. It demonstrates that the Fano shape analysis can be a tool to examine the local chemical environment. In the second type of film (phase II) a VO-up molecule showing a flat-lying bonding and a VO-down molecule with a tilted bonding appear in an alternative manner. The spin-polarized DFT calculation shows that, even though spin remains on all molecules, there are two types among VO-down molecules in terms of the spin coupling with neighboring molecules; first is that the spin has FM coupling with the flanking two flat-lying molecules and the other has AFM coupling. A clear Kondo peak appears at the former molecule but it disappears when measured at the latter. This phenomenon is explained by the competition between the Kondo resonance and RKKY interaction between the neighboring spins, which is enhanced with the presence of the reactive thiadiazole group.

## Methods

**Sample and film preparation**. VOTTDPz was synthesized following a previously reported method, which is described in detail in the our previous report[10]. We transferred the molecule to a substrate by using a sublimation method in an ultra-high vacuum (UHV). Substrate cleaning, molecule deposition, and low-temperature STM observations were carried out in UHV chambers, whose details are described elsewhere.[38,39]

**STM measurements**. The sample temperature was ~4.7 K for the STM/STS experiments described in this report. STS spectra were obtained using a lock-in amplifier with a modulation voltage of 1 mV superimposed onto the tunneling bias voltage.

**DFT calculation**. First-principle calculations were performed by using VASP code, employing a plane-wave basis set and projector augmented wave potentials to describe the behavior of the valence electrons.[40,41] A generalized gradient Perdew–Burke–Ernzerhof exchange-correlation potential was used[42]. The

structures were relaxed until the forces were smaller than 0.03 eV/Å. Due to the absence of dispersion forces in the local and semi-local exchange-correlation approximations, the molecule–surface distance of a weak bonding case such as van der Waals interaction is still controversial, which is followed by an ambiguity of the charge transfer from the substrate to the molecule. Nevertheless, the calculation results for the adsorbed molecule with van der Waals interactions gives an accuracy good enough to understand the spin behavior if compared with the result calculated for the molecules placed in the vacuum. The gold surface was modeled as a three-atom thick slab and the atoms in the bottom layer are fixed at the bulk position during the structure optimization. The Tersoff and Hamman theory[43] was used to compute the STM images using a previously reported method.[44] The calculations were performed on Numerical Materials Simulator at NIMS, Japan.

## Data availability
The data that support the findings of this study are available from the corresponding author upon request.

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

## Acknowledgements
This study was supported in part by Grantin-Aid for Scientific Research (S) (No. 19H05621 for T.K.), Scientific Research (S) (No. 16H06353 for K.A.), Scientific Research on Innovative Areas (No. 18H04482 for K.A.), and Scientific Research (B) (No. 15K13506 for A.R.). T.K. acknowleges financial supports by the Center for Spintronics Research Network (CSRN). R.A. acknowledges significant financial supports from the World Premier International Research Center Initiative (WPI) on Materials Nanoarchitectonics (MANA).

## Author contributions
J.H. and Y.W. performed S.T.M. K.E., C.N., and K.A. synthesized and characterized the VOTTDPz molecule. J.H., T.T., Y.S., and T.K. analyzed STM/STS data, R.A. performed DFT calculation. The manuscript was prepared by T.K. and J.H., and was approved by all authors.

## Competing interests

The authors declare no competing interests.
