## [Peer Review File · Communications Chemistry]

Reviewers' comments:

Reviewer #1 (Remarks to the Author):

Electronic spin coupling plays a fundamental role in magnetic materials, which leads to ferromagnetic and antiferromagnetic interaction. Studying molecular spin-spin coupling at the single-molecule level could help us to better understand organic magnetic materials. The manuscript by Jie Hou et al. reports spin-spin interaction of VOTDPz molecules on the Au(111) surface. By replacing the inert phthalocyanine ligand with reactive thiadiazole moiety, which has a similar shape to phthalocyanine but the benzene ring is terminated by S and N atoms instead of C-H, stronger spin-spin interaction compared to that for phthalocyanine molecules was realized. This was thoroughly investigated by a combination of scanning tunneling microscopy in a magnetic fields and density functional theory. The well-written manuscript presents a comprehensive story. I recommend to publish the nice paper after considering the following comments.

"First type of the film, phase I, is composed of flat lying molecules with the VO group directing towards the vacuum and with a more compact lattice compared to that of VOPc. For the spin property, we observed a significant change of spectrum-shape of the Kondo resonance when the tip position is moved within a single molecule; crossover from a generic Kondo peak to Fano dip, latter of which is caused by the interference of the tunneling currents. This is attributed to a local bonding variation within a single molecule; presence/absence of S-Au bonding among four thiadiazole ligands. This demonstrates that the shape variation of the Kondo resonance contributes to chemical analysis with sub-molecular resolution."

a. For Magnetic field dependent dI/dV curves of Kondo resonance shown in Fig. 5, why was the Zeeman effect not observed?

b. In Fig. 6, spectra were only measured at four positions, is it possible to measure more positions to get more details about position-dependent spin properties?

Reviewer #2 (Remarks to the Author):

In this manuscript, Jie Hou et al. report a new kind of magnetic molecules self-assemble on Au(111) surface in two phases. Low temperature scanning tunneling microscopy and spectroscopy show a spatial dependent Kondo signal within a single molecule in phase I and a reduced (pronounced) Kondo signal in the tilted molecule named b2(b1) in phase II. All the observations fulfill the understanding the behavior of magnetic molecules absorbed on metal surface, especially the related Kondo or RKKY effect. To this point, this paper is a good example to exhibit the complexity of such a system which may attract the interest of researchers working on this area. However, some important issues remain unclear and are critical to the explication of the experimental data. More data and discussions are still needed for me to make a decision. Some questions and comments are listed below:

1. In phase I, the molecules always show a cross like feature similar to that of MnPc molecule as shown in Figure 2. However DFT calculation done by the authors shows the molecule absorbed on Au(111) in this phase got a saddle-shaped configuration. I notice in a recent paper Nano Lett., 2017, 17 (8), pp 4929–4933, the molecule get the same configuration but only show a twofold symmetry in the STM topography image rather than fourfold symmetry reported here. The fourfold symmetry of the molecule here reflects its flat-lying feature just like MnPc on Au(111). The STM image do not agree with its DFT calculation, please make some comments on this point.

2. In Figure 3, the self-assembly model the author proposed in phase II is valid in logic. It will be better to give more experimental evidence to verify the tilted structure of b1 and b2 molecule. For example, the author can try to release the tilted molecule by pushing the other molecules surrounding it away using STM manipulation. If the tilted molecule can show a fourfold symmetry

after it becoming freestanding then the model the author proposed is very reasonable. The main concern here is whether b2 and b1 is still VOTTDpz, maybe some new molecule after surface reaction or even maybe some impurity. I notice the authors have optimized their proposed model using DFT calculation, why not trying some STM simulation using DFT to make a comparison with the image obtained in experiment. By the way, the d1 and d2 molecule in the STM image seems to have some deformation from the cross feature in phase I, which may indicate the molecule experience some strain or interaction from its surrounding. Whether this can lead some effects on the STS in Figure 7?

3. Field dependent STS in Figure 5 is the most confusing data to me. Although the step-like feature to a sharp peak around zero energy is similar to what Liu (PRL 2015) observed when MnPc was dehydrogenated at 0T. However, in Liu's paper the Kondo effect was verified by peak splitting with increasing magnetic field. But in this paper, there is no direct evidence shows this is really a Kondo effect. I understand the splitting is hard to be seen in an experiment condition of 4K and 5T. The authors can lead some further experiment (Low temperature and high field) to verify their idea (magnetic field induced charge transfer). To my point of view, this is not a simple spin 1/2 Kondo effect. Some other model can also be discussed (like Kondo + mixed valence) to fit the experimental data.

4. The author attribute peaks around -40 mV and -20 mV in Figure 7 to vibration mode. I notice these peaks in d1 are more pronounced than b1. However, d1 is in a flat configuration similar to that in phase I. This looks quite controversy to the explanation the author mentioned in the main text in which they claim vibration benefit more from tilted configuration.

In summary, this paper reported the complex interaction in magnetic molecules but some in depth analysis of the data are still needed. One more comment, there are 8 figures in the main text. However, I think they could be rearranged and be more compact.

Reviewer #3 (Remarks to the Author):

Hou et al present an interesting piece of work with many data on an interesting system. The use of substituted phthalocyanine molecules is very interesting and the authors have shown success in studying this system in previous publications. The authors study the Kondo peak appearing in tunneling dI/dV spectra and from the variation of the Kondo peak within the molecule and with respect to other molecules in the same layer, they conclude on the spin ordering of the molecules. Unfortunately, a few assumptions and unproved conclusions render the work difficult to assess. Particularly, it would be beneficial to have some independent measurement on the spin ordering to be able to claim that the change in Kondo shape reflects the magnetic ordering of the molecular layer. Can the authors provide more data or complementary data showing that this is indeed the case? All in all I think this work deserves publication, but I am not sure whether the main analysis tool is tested enough to render all conclusions valid.

Other issues with the paper:

1. Can the authors explain why the Kondo peak appears at -6 meV at 0T for phase I?
2. Any reason why the Kondo peak shifts to zero as the magnetic field is ramped? Why does it get thinner? My impression is that the magnetic field is inducing some mechanical distortion of the system and the coupling of the molecules with the substrate is reduced...otherwise the magnetic field dependence does not seem to prove this is a Kondo system.
3. What is the direct molecule-molecule interaction? I would say that there is direct exchange interaction between molecules and the RKKY interaction is not needed to explain the experiment. Can the authors make an estimation of order of magnitude? I would expect to find direct exchange in the meV range while RKKY in the micro-eV one.

In summary, I recommend publication after revision of the above issues by the authors.

Reviewer #1

a. For Magnetic field dependent dI/dV curves of Kondo resonance shown in Fig. 5, why was the Zeeman effect not observed?

Thank you for your precious comments.

We understand the referee's point concerning the importance of the Zeeman split in the presence of the magnetic field, which is a powerful method to examine whether the feature is derived from the Kondo resonance or not.

However, mainly due to the relatively wide peak width of the Kondo feature of this system, the splitting cannot be clearly seen. The situation is now explained in detail like following in the main text.

" One might wonder why the Zeeman splitting is not obvious for the spectra with the application of $B=5$ T. Park and coworkers demonstrated that, when outer-magnetic field is applied, the Kondo peak is split into two peaks with the energy separation of twice the Zeeman energy³¹. However, compared to this case, the original broadening of the Kondo feature is larger in the experiment shown in this report which makes it difficult to see the splitting. In our previous report, we simulated the changes in the peak by convoluting two Lorentzian peaks with the width of 6.5 meV and separated by 1.8 meV (two times the Zeeman splitting energy at 8 T) around the Fermi level into a single peak. Due to the original peak width of ~ 6.5 mV, the splitting was not obvious except for a slight broadening of ~ 1 meV. Since a similar broadening in the original peak, the Zeeman splitting is not obvious even with the application of the outer magnetic field."

b. In Fig. 6, spectra were only measured at four positions, is it possible to measure more positions to get more details about position-dependent spin properties?

We have added more spectra in Figure 6, which makes the comparison between the inner position of the molecule and the Kondo feature easier. Thank you.

Reviewer 2

Thank you for your precious comments.

In phase I, the molecules always show a cross like feature similar to that of MnPc molecule as shown in Figure 2.

However DFT calculation done by the authors shows the molecule absorbed on Au(111) in this phase got a saddle-shaped configuration.

I notice in a recent paper Nano Lett., 2017, 17 (8), pp 4929–4933, the molecule get the same configuration but only show a twofold symmetry in the STM topography image rather than fourfold symmetry reported here.

The fourfold symmetry of the molecule here reflects its flat-lying feature just like MnPc on Au(111). The STM image do not agree with its DFT calculation, please make some comments on this point.

We re-examine the STM data, following the referee's comment. It was revealed that the height profiles measured along two symmetry lines are not equivalent. This explains the results of our DFT calculation well, and it is likely the topographic image of the molecule shows two-fold symmetry. We add new figures in Fig. 2(d).

The following section is newly added.

"We can detect this bent configuration of the molecule in the topo image by measuring the cross-sectional height variation in the lateral position. In Figure 2(d) we show the cross-sectional height information of the molecule in the phase I film along the two symmetry lines, which are indicated by the colors of red and black superimposed on the topographic image. The height profile illustrated in Figure 2(d) suggests that the heights of the lobes along these two lines are not identical. Rather, the one along the red line is bent towards the vacuum side compare to that along the black. Thus, it is in the two-fold symmetry condition even the difference is rather small. This supports the optimized structure shown in Figure 2(c) in which the bending angles of the lobes along the two symmetry lines are not identical."

In Figure 3, the self-assembly model the author proposed in phase II is valid in logic. It will be better to give more experimental evidence to verify the tilted structure of b1 and b2 molecule. For example, the author can try to release the tilted molecule by pushing the other molecules surrounding it away using STM manipulation. If the tilted

molecule can show a fourfold symmetry after it becoming freestanding then the model the author proposed is very reasonable.

Thank you for your comments. The idea of the referee of the use of the manipulation technique to check the bonding configuration is very intriguing. We have tried the manipulation. However, it was not well executed and we cannot show the results as the evidence of the canted bonding.

The main concern here is whether b2 and b1 is still VOTDPz, maybe some new molecule after surface reaction or even maybe some impurity.

We understand the concern of the referee that these features can be originated from impurities. However, as shown in Figure 1, the number of the molecules included in phase II is as large as that of phase I. The purity of the original sample molecules is quite good and it is not possible to consider the phase II is composed of impurity molecules. For the possibility of the chemical reaction on the surface, we believe the Au(111) is inert if all experiments were done at temperature below RT.

I notice the authors have optimized their proposed model using DFT calculation, why not trying some STM simulation using DFT to make a comparison with the image obtained in experiment.

As the referee kindly suggested, we provide the STM simulation based on the DFT calculation is provided as a new figure in Figure 3.

By the way, the d1 and d2 molecule in the STM image seems to have some deformation from the cross feature in phase I molecule experience some strain or interaction from its can lead some effects on the STS in Figure 7?

We carefully compare the topo image of the d1 and d2 with those of Figure 2, which shows a certain difference between them.

However d1 and d2 molecules are more overlapped with b1 and b2 of neighbor molecules

Thus the difference of the STM images cannot be attributed to the effect of the strain. We discuss the magnetic interaction between the neighboring molecules but interaction between the neighboring molecules cannot be well separated.

Thank you for your precious comments.

Field dependent STS in Figure 5 is the most confusing data to me. Although the step-like feature to a sharp peak around zero energy is similar to what Liu (PRL 2015) observed when MnPc was dehydrogenated at 0T. However, in Liu's paper the Kondo effect was verified by peak splitting with increasing magnetic field. But in this paper, there is no direct evidence shows this is really a Kondo effect. I understand the splitting is hard to be seen in an experiment condition of 4K and 5T.

As previous reports demonstrated, the Kondo peak is expected be split into two peaks by applying the outer magnetic field, whose energy separation is twice of the Zeeman energy. However, it is difficult to observe this split in our system with the reasons as the referee kindly suggested. Since two other referees raised a similar question about the absence of the Zeeman splitting, we added the following section as the detail of the behavior of the ZBP with the outer magnetic field.

" One might wonder why the Zeeman splitting is not obvious for the spectra with the application of $B=5$ T. Park and coworkers demonstrated that, when outer-magnetic field is applied, the Kondo peak is split into two peaks with the energy separation of twice the Zeeman energy³¹. However, compared to this case, the original broadening of the Kondo feature is larger in the experiment shown in this report which makes it difficult to see the splitting. In our previous report, we simulated the changes in the peak by convoluting two Lorentzian peaks with the width of 6.5 meV and separated by 1.8 meV (two times the Zeeman splitting energy at 8 T) around the Fermi level into a single peak. Due to the original peak width of ~ 6.5 mV, the splitting was not obvious except for a slight broadening of ~ 1 meV. Since a similar broadening in the original peak, the Zeeman splitting is not obvious even with the application of the outer magnetic field."

The authors can lead some further experiment (Low temperature and high field) to verify their idea (magnetic field induced charge transfer).

Thank you for your comments. Unfortunately we cannot execute the experiment below 1 K.

To my point of view, this is not a simple spin 1/2 Kondo effect. Some other model can also be discussed (like Kondo + mixed valence) to fit the experimental data.

We introduced in the main text that the previously reported Kondo features for the phthalocyanine molecules can be divided into two groups and MnPc molecule is listed as an example of the peak with an asymmetric peak shape. However, the report for the MnPc molecule by Liu and coworkers shows that the energy position is at the Fermi level, which shows a significant difference from our case shown in this paper.

Thus, after examining and analyzing the similarity and the discrepancy from that work, we discussed the possibility of the mixed valence state that caused the complex shape observed for the smaller magnetic.

Mixed valence is well described in the newly added paragraph like following.

“Instead, the change of the ZBP can be accounted for with a model of the cross-over from the Kondo regime to the mixed valence state (MV). With the Anderson model, the Kondo resonance is modeled with a singly occupied electronic state. Several regimes are considered depending on the energy position E_0 and the peak width Δ of the singly occupied state; when $E_0/\Delta \gg 1$ it is grouped as the Kondo state, while the broadened state is overlapped with the Fermi level ($E_0/\Delta < 1$) it is grouped as the mixed valence state.

The spectral functions, which can simulate the tunneling spectra, are calculated theoretically in several reports. Li and coworkers calculated a case where the magnetic atom is placed on the graphene substrate.²⁷ Also, we can see similar cross-over simulation for the rare earth inter-metallics²⁸, together with an experimental report²⁹.

In the Kondo regime, the spectral function shows a peak located at the Fermi level with a sharp peak width, while in the MV regime, the peak is off the Fermi level and the peak width is wide. The former and latter features are what were observed for $B=5$ T and $B=0$ T in this experiment, respectively. Thus, we consider that a crossover from the MV regime to the Kondo resonance regime is induced by the magnetic field. An example of the switching from the Kondo region to the mixed valence regime can be seen in the CNT FET device. By tuning the gate voltage, the relative position of the spin center in reference to the Fermi level can be tuned. The results showed characteristic features of the conductance in both regimes³⁰.”

The author attribute peaks around -40 mV and mode. In 20 mV in Figure 7 to vibration otice these peaks in d1 are more pronounced than b1. However, d1 is in a flat configuration similar to that in phase I . This looks quite controversy to the

explanation the author mentioned in the main text in which they claim vibration benefit more from tilted configuration.

Thank you for your precious comments.

Our previous discussions were based on many speculations. Following to the referee's comments, we deleted to the attribution to the phonon mode.

Reviewer 3

Hou et al present an interesting piece of work with many data on an interesting system. The use of substituted phthalocyanine molecules is very interesting and the authors have shown success in studying this system in previous publications. The authors study the Kondo peak appearing in tunneling dI/dV spectra and from the variation of the Kondo peak within the molecule and with respect to other molecules in the same layer, they conclude on the spin ordering of the molecules.

Unfortunately, a few assumptions and unproved conclusions render the work difficult to assess. Particularly, it would be beneficial to have some independent measurement on the spin ordering to be able to claim that the change in Kondo shape reflects the magnetic ordering of the molecular layer.

Can the authors provide more data or complementary data showing that this is indeed the case? All in all I think this work deserves publication, but I am not sure whether the main analysis tool is tested enough to render all conclusions valid.

Thank you for your precious comments.

Since it has been demonstrated that X-ray magnetic circular dichroism (XMCD) is the most adequate technique, we added our preliminary data of XMCD in Supporting Information. Compared with the VOPc case, the measured magnetization for the system discussed in this manuscript is smaller. We consider that this can be a supporting material for the AFM coupling between the molecules.

We added the following sections in the main text.

"We discuss an additional technique to verify such an FM/AFM magnetic ordering. It has been demonstrated that the X-ray magnetic circular dichroism (XMCD) is a powerful technique to analyze the magnetic behavior of the molecule film. We show preliminary result of the XMCD comparing the films of the VOPc and VOTTDPz molecules in Supporting Information (see S3 of Supporting Information). The XMCD results indicate that the magnetization appeared on the VOTTDPz film is much weaker than that obtained from the VOPc film. The result can be interpreted by a model that, compared to the paramagnet nature of VOPc, the appearance of the AFM magnetic ordering between molecules in a part of the film (corresponding to the phase II part) should correspond to the decrease of the magnetization."

In addition, the following is added in the Supporting Information.

"S3 XMCD measurement on VOPc and VOTTDpZ

Figure S2. Circular polarized XAS of films of VOPc (a), and TbPc2 (b) on Au(111) at T=5 K and B=5 T at 0° (normal) and 55° incidence (I+, I-) and XMCD for each case. Note XMCD plot for VOTTDpZ is doubled in y-scale.

We compare the circular polarized x-ray absorption spectroscopy (XAS) obtained on the films of VO phthalocyanine (VOPc) and VOTTDpZ molecules on Au(111). In each panel, XAS results for two circular directions and the XMCD plots are illustrated. The photon energy range includes V 2p and O 1s components. The XMCD intensity of the VOTTDpZ component is doubled for the clarification. The XMCD intensity comparison between the VOPc film and the VOTTDpZ film, by considering the doubled scale for the VOTTDpZ film, indicates the magnetization is weaker for the VOTTDpZ film."

Other issues with the paper:

1. Can the authors explain why the Kondo peak appears at -6 meV at 0T for phase I?
2. Any reason why the Kondo peak shifts to zero as the magnetic field is ramped? Why does it get thinner? My impression is that the magnetic field is inducing some mechanical distortion of the system and the coupling of the molecules with the substrate is reduced...otherwise the magnetic field dependence does not seem to prove this is a Kondo system.

The following section is added in the main text.

"Instead, the change of the ZBP can be accounted for with a model of the cross-over from the Kondo regime to the mixed valence state (MV). With the Anderson model, the Kondo resonance is modeled with a singly occupied electronic state. Several regimes are considered depending on the energy position E_0 and the peak width Δ of the singly

occupied state; when $E_0/\Delta \gg 1$ it is grouped as the Kondo state, while the broadened state is overlapped with the Fermi level ($E_0/\Delta < 1$) it is grouped as the mixed valence state.

The spectral functions, which can simulate the tunneling spectra, are calculated theoretically in several reports. Li and coworkers calculated a case where the magnetic atom is placed on the graphene substrate.²⁷ Also, we can see similar cross-over simulation for the rare earth inter-metallics²⁸, together with an experimental report²⁹.

In the Kondo regime, the spectral function shows a peak located at the Fermi level with a sharp peak width, while in the MV regime, the peak is off the Fermi level and the peak width is wide. The former and latter features are what were observed for $B=5$ T and $B=0$ T in this experiment, respectively. Thus, we consider that a crossover from the MV regime to the Kondo resonance regime is induced by the magnetic field. An example of the switching from the Kondo region to the mixed valence regime can be seen in the CNT FET device. By tuning the gate voltage, the relative position of the spin center in reference to the Fermi level can be tuned. The results showed characteristic features of the conductance in both regimes³⁰.

For the mechanism of the crossover from the MV regimes to the Kondo regime with B application, we should consider the mechanical distortion of the molecule with the presence of the magnetic field, which can induce the reduction of the coupling between the molecule and the substrate. However, much more detail should be examined in the future."

3. What is the direct molecule-molecule interaction? I would say that there is direct exchange interaction between molecules and the RKKY interaction is not needed to explain the experiment. Can the authors make an estimation of order of magnitude? I would expect to find direct exchange in the meV range while RKKY in the micro-eV one.

Thank you for your comments. We understand the concern of the referee.

RKKY is J^2 effect and is small in the magnitude. As can be seen in the reference 16, Tsukahara et al. experimentally showed that it is less than 1 meV, which is much smaller than the expected direct molecule-molecule interaction. However, the point we like to stress is, even though much difference was observed for the Kondo feature measured at b1 and b2, the local bonding configuration with the neighboring molecules of d1 and d2 are same for b1 and b2 molecules. This is added in the main text like following, which explains the necessity to introduce the model of RKKY interaction.

"In addition, a direct interaction of the electronic states between the neighboring

molecules can cause such an effect. However, it should be noted that the two molecules of b1 and b2 are equivalent in terms of the local structure including the neighboring molecules of d1 and d2, which should be followed by the identical molecule-molecule interaction. Thus, the presence and the absence of the Kondo state at b1 and b2 cannot be explained solely by this effect."

REVIEWERS' COMMENTS:

Reviewer #1 (Remarks to the Author):

I have read the revised manuscript and rebuttal letter carefully. It can be polished without further revision.

Reviewer #2 (Remarks to the Author):

In the revised manuscript, the authors improved both their figures and analysis a lot. They answered most of my questions provided in previous report.

However, there is still one point need to be addressed. The line profile in Figure 2d is not correct. In the upper panel of figure 2d, the red line crosses the black line. The cross point is named as center if I understand correctly. Therefore, in the lower panel of figure 2d, the red line and the black line should have a same height at the center. But it is not.

After they address this point, I think the manuscript can be published in the journal.

Reviewer #3 (Remarks to the Author):

I think the authors have satisfactorily answered all queries rised by all referees. I recommend publication as is.

Reply to Reviewer #1 (Remarks to the Author):

I have read the revised manuscript and rebuttal letter carefully. It can be polished without further revision.

Thank you very much. We really appreciate your efforts.

Reply to Reviewer # 2 (Remarks to the Author):

In the revised manuscript, the authors improved both their figures and analysis a lot. They answered most of my questions provided in previous report.

However, there is still one point need to be addressed. The line profile in Figure 2d is not correct. In the upper panel of figure 2d, the red line crosses the black line. The cross point is named as center if I understand correctly. Therefore, in the lower panel of figure 2d, the red line and the black line should have a same height at the center. But it is not.

After they address this point, I think the manuscript can be published in the journal.

Thank you for your precious comment. We understand your point correctly. The graphe of Fig. 2d is now revised so that the two lines of the height-profiles coincide at the center point. We made a careless mistake to make the reference point at the bottom of each curves. We apologize for making confusion. We also appreciate all your efforts to improve our manuscript.

Reply to Reviewer #3 (Remarks to the Author):

I think the authors have satisfactorily answered all queries raised by all referees. I recommend publication as is.

Thank you for all your efforts for the evaluation of our manuscript.